# Development and psychometric properties of maternal health literacy inventory in pregnancy

Safoura Taheri[1], Mahmoud Tavousi[2], Zohre Momenimovahed[3], Ashraf Direkvand-Moghadam[1], Azita Tiznobaik[4], Zainab Suhrabi[1], Ziba Taghizadeh[5]*

**1** Department of Midwifery, School of Nursing & Midwifery, Ilam University of Medical Sciences, Ilam, Iran,
**2** Health Metrics Research Center, Iranian Institute for Health Sciences Research, ACECR, Tehran, Iran,
**3** Department of Midwifery, School of Nursing and Midwifery, Qom University of Medical Sciences, Qom, Iran, **4** Department of Midwifery, Faculty of Nursing and Midwifery, Member of Mother and Child Care Research Center, Hamadan University of Medical Sciences and Health Services, Hamadan, Iran,
**5** Department of Midwifery and Reproductive Health, School of Nursing and Midwifery, Tehran University of Medical Sciences, Tehran, Iran

* zibataghizadeh@yahoo.com

**Data Availability Statement:** All relevant data are within the paper and its Supporting Information files.

## Abstract

### Background

Pregnancy is one of the most sensitive and important stages of women's life. Maternal health literacy is the key to achieving a healthy pregnancy. It also affects pregnancy outcomes by improving the quality of health care in this period. The aim of this study was to develop and evaluate the psychometric properties of maternal health literacy inventory in pregnancy (MHELIP).

### Methods

This sequential, exploratory and mixed study was carried out in two parts (qualitative study and psychometric evaluation of the tool) in Tehran in 2016–18. The first part involved a qualitative content analysis with a traditional approach using in-depth, semi-structured and personal interviews with 19 eligible pregnant women. Then, the pool of items extracted from the qualitative part was completed by reviewing the existing literature and tools. In the second part, the overlapping items were summarized and the tool was validated. In order to evaluate the construct validity, a cross-sectional study was conducted with the participation of 320 pregnant women. Data analysis was performed by SPSS-19 software using exploratory factor analysis and reliability tests (Cronbach's alpha and ICC).

### Results

Findings of qualitative part produced a pool of 120 items that reached to 124 items after reviewing the literature. After confirming face and content validity by calculating CVI and CVR for each item, 53 items remained in the pool. Finally, the results of exploratory factor analysis confirmed a tool with 48 items in four factors, explaining 46.49% of the variance of total variables of the tool. Reliability of the tool was approved by Cronbach's alpha = 0.94

**Funding:** The authors received no specific funding for this work.

**Competing interests:** The authors have declared that no competing interests exist.

and test-retest with 2-weeks intervals, indicating an appropriate stability for the scale (ICC = 0.96). Finally, the tool was finalized with 48 items in 4 dimensions, including "Maternal Health Knowledge", "Maternal Health Information Search", "Maternal Health Information Assessment" and "Maternal Health Decision Making and Behavior".

## Conclusion

The designed tool is a multidimensional, reliable and validated scale for assessing maternal health literacy during pregnancy. This tool can be used to evaluate different aspects of maternal health literacy in pregnant women, which was prepared based on their experiences during a qualitative study.

## Introduction

Pregnancy is one of the most sensitive and important stages of women's life. The importance of pregnancy is remarkable as the health and well-being of mother directly affect the current and future lives of the fetus [1, 2]. According to the World Health Organization's (WHO) 2010 report, approximately 1,000 pregnant women worldwide die every day due to pregnancy and childbirth complications, with 99% of them occurring in developing countries [3].

Early onset of prenatal care in the first trimester of pregnancy and its continuation during pregnancy will lead to improved pregnancy and delivery outcomes. Careful prenatal care will prevent maternal-fetal complications [4]. However, despite the provision of different forms of care by health care providers, some factors appear to prevent the proper and timely delivery of care during pregnancy. One of these factors is the lack of maternal health literacy during pregnancy [5].

Maternal health literacy refers to cognitive and social skills that motivate and enable women to acquire, understand, and use information in a way that safeguards and promotes their health and health of their children [6]. Health literacy is the key to achieving a healthy motherhood and has an impact on pregnancy outcomes by improving the quality of health care during pregnancy [7]. Maternal health literacy empowers women to receive timely prenatal, decision making, and labor management education such as accepting midwifery interventions and even pain management [8, 9]. In a qualitative study that examined the understanding and perception of pregnant women about health literacy (pregnancy risk symptoms, preparation for childbirth and its complications, and understanding of neonatal care), the results showed that women were aware of the risks of pregnancy, but they could not properly interpret and apply the key information provided during prenatal care [10]. In a study in Tanzania, 42% of pregnant women [11] and in a study conducted by Haun et al (2014), 30% of mothers did not know any of the signs of risk in pregnancy and childbirth [12].

It should be noted that, because health literacy has a variety of definitions and structures, researchers have designed different tools for measuring health literacy in different groups, and each tool measures one aspect of health literacy. A number of known tools include oral health literacy [13], adolescent health literacy [14], health literacy in women with breast cancer [15], health literacy in people with chronic diseases such as diabetes [16, 17], Asthma [18] and hypertension [19], health literacy competencies [20], scale of low salt consumption [21], and weight literacy scale [22]. Abel (2014) states that, examining health literacy in different domains requires different research approaches [23]. On the one hand, it is obvious that health literacy has different meanings and dimensions in different periods of life and cannot be

generalized by the tools that have medical terminology and information on general diagnostic and therapeutic methods.

Knowing about the level of health literacy is essential to properly address the information needed by pregnant women [24]. Studies of pregnant women's health literacy have been using general health literacy tools that have reported different results regarding the level of pregnant women's health literacy [25–30]. Some studies have also examined the health literacy level of pregnant women using a researcher-made questionnaire and reported that only 18 to 24% of pregnant women had good health literacy and 31 to 34% of them had poor health literacy [8, 31]. Charoghchian Khorasani et al (2017) used the maternal health literacy and pregnancy outcome questionnaire designed by Mojoyinola et al (2011) and reported that, the level of health literacy in Iranian pregnant women is at a low level [32, 33]. By examining the level of health literacy, especially at the beginning of prenatal care, we can identify those who have low levels of maternal health literacy, identify geographical areas with low maternal health literacy, and plan effective interventions in this regard.

According to search conducted in the scientific literature in the world, we found several studies on tool design to measure maternal health literacy [33–35], but a native tool derived from an original research with tool design methodology (including the use of qualitative method that determines this concept from the perspective of pregnant mothers as the target group) could not be found. Available studies have used general tools or researcher-made questionnaires to determine the level of maternal health literacy during pregnancy, which seem to have only assessed the health knowledge rather than health literacy. It has been emphasized that health literacy goes beyond the health knowledge [36], but is closely linked to health knowledge [37, 38]. Therefore, this study aimed to design and psychometrically evaluate a maternal health literacy tool during pregnancy.

## Material and methods

The present study is a sequential, exploratory and mixed research that was conducted between November 2016 to August 2018. This research started with a qualitative study and continued with a quantitative study. This method was introduced by Creswell and Plano Clark as one of the five main types of mix method studies [39]. The sequential exploratory study is divided into two types of theory design and instrument design. The present study is a tool design study [40]. To build a tool, we need to understand the concept of health literacy by using the experience of participants. Having a conceptual framework is the first step in designing a tool. Therefore, the qualitative part of the study was designed and implemented to design the tool, and then its psychometric property was confirmed in a quantitative study.

### Step one: Qualitative study

The first phase of the study was performed from February 2016 to September 2017. In the qualitative phase, we conducted in-depth, semi-structured and personal interviews with 19 pregnant women referred to medical and health center affiliated to Tehran University of Medical Sciences. Inclusion criteria were; being a pregnant woman with minimum level of education (reading and writing) and being an Iranian citizen. Interviews were conducted at the locations agreed upon by the researcher and participants (such as prenatal education classrooms in hospitals or comprehensive health centers, parks near the participants' place of residence, etc.) The duration of each interview was between 30 to 130 minutes with an average of 60 minutes, depending on the amount of information, participation, and cooperation of the participants. Qualitative analysis was performed using conventional content analysis through MAXQDA-10 software. After analyzing the findings of qualitative section and reviewing the

available literature and tools, a 124-item pool was created. After reviewing the items, the members of primary research group developed a 78-item Maternal Health Literacy Assessment Tool (MHELIP) questionnaire, consisting of two sections: 1) Assessing the information related to pregnancy health, and 2) Functional health literacy. For the initial scoring, a 5-point Likert scale was used in the section of assessing the information related to pregnancy health, which ranged from I don't know at all to I know a lot (with the score of 1 to 5), and a scale of never to always (with the score of 1 to 5) was used in the section of functional health literacy. Then, the research entered the quantitative phase and psychometric evaluation of the designed tool was performed.

## Step two: Tool's psychometric

In the second stage of this study, to check the validity and reliability of the tool, face validity, content validity, construct validity and reliability (internal consistency and stability) were used. These stages of the study were performed between October 2017 and June 2018.

**1. Validity.**  *Content validity*. In order to evaluate the qualitative content validity, 10 experts in gynecology, midwifery, reproductive health, maternal and child health, health education, nursing, and health literacy were asked to review the tool and express their comments and opinions in terms of its grammar and the use of right words and phrases, and also offer suggestion to add or remove items in writing. The questionnaire was then edited according to the experts' recommendations. Content validity ratio (CVR) and content validity index (CVI) were used to assess the content validity of the tool. In order to calculate the content validity ratio (CVR), fourteen specialists were asked to assess each item in a 3-point Likert scale (essential = 2, useful but not essential = 1 and not essential = 0). Then, based on Lawshe's table [41], items that received the score of 0.51 or above were kept in the tool. In order to calculate the Content Validity Index (CVI), 14 specialists were asked to determine the relevance, clarity, and simplicity of each item, using a 4-point Likert scale. According to Waltz and Baussel [42], the items with CVI value of 0.79 or above were accepted.

*Face validity*. In the qualitative face validity, the difficulty of understanding the items, words and the ambiguities, and also the possibility of misinterpreting the items or the words were assessed after interviewing 15 pregnant women referred to medical and health centers in Tehran. During this phase, the impact score of each item was calculated (impact score = frequency (%) × importance). For this purpose, another sample of 15 pregnant women referred to medical and health centers in Tehran were asked to score the importance of each item based on a 5-point Likert scale. The items with impact scores of 1.5 or above were considered to be satisfactory [43]. Following the content validity and face validity assessment, the pre-final version of the instrument (maternal health literacy inventory in pregnancy-MHELIP) had 53 items and was ready for the quantitative phase (Table 1).

**2. Construct validity.**  In the present study, before determining construct validity, the initial reliability (correlation between items) was assessed. Reliability of the original questionnaire was confirmed by conducting convenience sampling among 30 pregnant women referred to

**Table 1. Steps of reducing the item from the initial pool of items to the final version of the questionnaire.**

| Stage | Items | Results | Number of remaining items |
|---|---|---|---|
| Primary pool | 128 | Remove and integration of the 46 items in several meetings | 78 |
| Content validity (qualitative and quantitative) | 78 | Remove and integration of the 25 items | 53 |
| Face validity (quantitative and qualitative) | 53 | - | 53 |
| Exploratory factor analysis | 53 | Remove 5 items and transfer 2 items | 48 |
| Reliability | 48 | - | 48 |

medical and health centers of Tehran with Cronbach's alpha coefficient of 0.94 and as a result, the tool with high reliability entered into the construct validity stage. Exploratory factor analysis approach was used for construct validity. In this phase, 320 pregnant women referred to medical and health centers of Tehran completed the maternal health literacy questionnaire. Inclusion criteria were; Ability to read and write, having Iranian citizenship, having no history of medical education, and giving consent to participate in the study. The exclusion criteria included; not completing the questionnaire fully and having high level of stress in the last six months such as losing loved ones, etc. First, the sampling adequacy test of KMO and the Bartlett's test were used, and then the analysis of main items, varimax rotation, and factor analysis of the questionnaire were determined.

**3. Reliability.** After confirming the validity of the questionnaire, reliability was assessed by internal consistency and test-retest methods. In the internal consistency method, consistency of the results of the tool items was investigated and then, the Cronbach's alpha coefficient was calculated for the items in each domain and the whole questionnaire. In order to perform the test-retest, 20 pregnant women referred to medical and health centers of Tehran completed the final questionnaire within two weeks and the intraclass correlation coefficient (ICC) was calculated.

**4. Ethical statement.** The Ethics Committee of the Faculty of Nursing and Midwifery of Tehran University of Medical Sciences, consisting of an Institutional Review Board with the presence of participating experts reviewed and approved our study (Code of Ethics approved: IR.TUMS.VCR.REC.1395.1866). Participants provided informed written consent prior to participating in the research study. All participants were reminded that their participation is voluntary and they have the right to withdraw from the research at any stage. All personal data such as names were anonymized. Paper based data (consent) was stored securely in a locked cupboard and electronic data (interview and transcripts, quantitative data) was stored in secure server with password (The consent form used is attached in the S1 File section).

## Results

### 1. Qualitative study

In total, 19 pregnant women aged 16–45 years participated in this study, from whom 12 were nulliparous and 9 were multiparous. The lowest gestational age was 7 weeks and the highest gestational age was 39 weeks. The participants' level of education was between secondary school and PhD student and they were mostly housewives. After analyzing the findings of qualitative section and reviewing the available literature and tools, a 124-item pool was created. After reviewing the items, the members of primary research group developed a 78-item Maternal Health Literacy Assessment Tool (MHELIP) questionnaire, consisting of two sections; 1) assessing the information related to pregnancy health, and 2) functional health literacy. For the initial scoring, a 5-point Likert scale was used in the section of assessing the information related to pregnancy health, which ranged from I don't know at all to I know a lot (with the score of 1 to 5), and a scale of never to always (with the score of 1 to 5) was used in the section of functional health literacy.

### 2. Content validity

The initial questionnaire of maternal health literacy during pregnancy had 78 items. In the qualitative content validity, the items were reviewed based on the opinions of 10 experts and necessary amendments were made to the items. At this stage, 24 items were removed. In the quantitative content validity, two indexes of content validity ratio and content validity index were calculated for 54 items. In the content validity ratio, according to the number of experts

(14 persons), the minimum accepted CVR was 0.51 and in the content validity index only one item "I can solve pregnancy problems" with a score of 0.14 was removed from the original questionnaire. Finally, the questionnaire with 53 questions obtained the construct validity.

### 3. Face validity

In the qualitative face validity, 15 pregnant women were asked to read and provide feedback on the questionnaire, and then the proposed corrections were applied but no item was removed. In the quantitative face validity, 15 pregnant women were asked to comment on the importance of each item to determine the impact factor of each item, and all items obtained the score of above 1.5 and therefore no item was removed in this stage.

### 4. Construct validity

The mean age of participants in this section of study was 28.32 ± 5.59 years and their mean gestational age was 26.75 ± 9.31 weeks. The majority of women (51.3%) were experiencing their first pregnancy and most of them (76.3%) had intended pregnancy. Most of the pregnant women had high school diploma (41.6%) and were housewife (94.1%). Most of the participants (45.6%) had the household income of between one and two million tomans and the majority of their spouses also had high school diploma (43.6%). Also, 44.4% of the participants had not attended any pregnancy training course and most of them (80.3%) had internet access.

In the first step, the adequacy of sampling was assessed for the factor analysis. The statistical value of KMO = 0.905 indicated the adequacy of sampling (sample size of 320 pregnant women) for factor analysis. Bartlett's test showed significant relationship between the items (P- value <0.001). Exploratory factor analysis was performed on 53 items in two stages. In the first stage, initial analysis of variables was performed using specific value of greater than 1 and varimax rotation, which explained 65.18% variance with 12 factors. Then, the Scree Plot was used to determine the number of factors and based on this chart, 4 factors were proposed for extraction (Fig 1). To ensure the structure of items, the 4 to 7-factor structure was tested with different rotations, and the best of them was identified to be the 4-factor structure. Exploratory factor analysis was performed with varimax rotation and 4 factors (Table 2). To keep the items, minimum load factor was set at 0.35. All items had the factor load of higher than 0.35. In the exploratory factor analysis phase, 5 items were removed from the maternal health literacy tool. Finally, after performing factor analysis, 4 factors with 48 items explained 46.49% of the total variance. The first factor with 21 questions explained 20.73% of the variance, second factor with 15 questions explained 12.81% of the variance, third factor with 6 questions explained 7.95% of the variance, and fourth factor with 6 questions explained 4.99% of the variance. Accordingly, the factors were named; "Maternal Health Knowledge", "Maternal Health Decision Making and Behavior", " Maternal Health Information Assessment" and "Maternal Health Information Search", respectively (Table 2). At the end, the questionnaire was completed with 48 items.

### 5. Reliability

To determine internal consistency, after ensuring construct validity, Cronbach's alpha coefficient in a sample of 320 pregnant women was 0.94 for the whole tool with 95% confidence intervals. However, this value varied from 0.66 to 0.94 in different dimensions. In order to determine the consistency of questionnaire in the repeatability dimension, in a group of 20 pregnant women referred to medical and health centers of Tehran with 2 weeks' interval, the intraclass correlation coefficient (ICC) for the whole tool was 0.96 with 95% confidence intervals, and this value varied from 0.74 to 0.97 in different dimensions (Table 3).

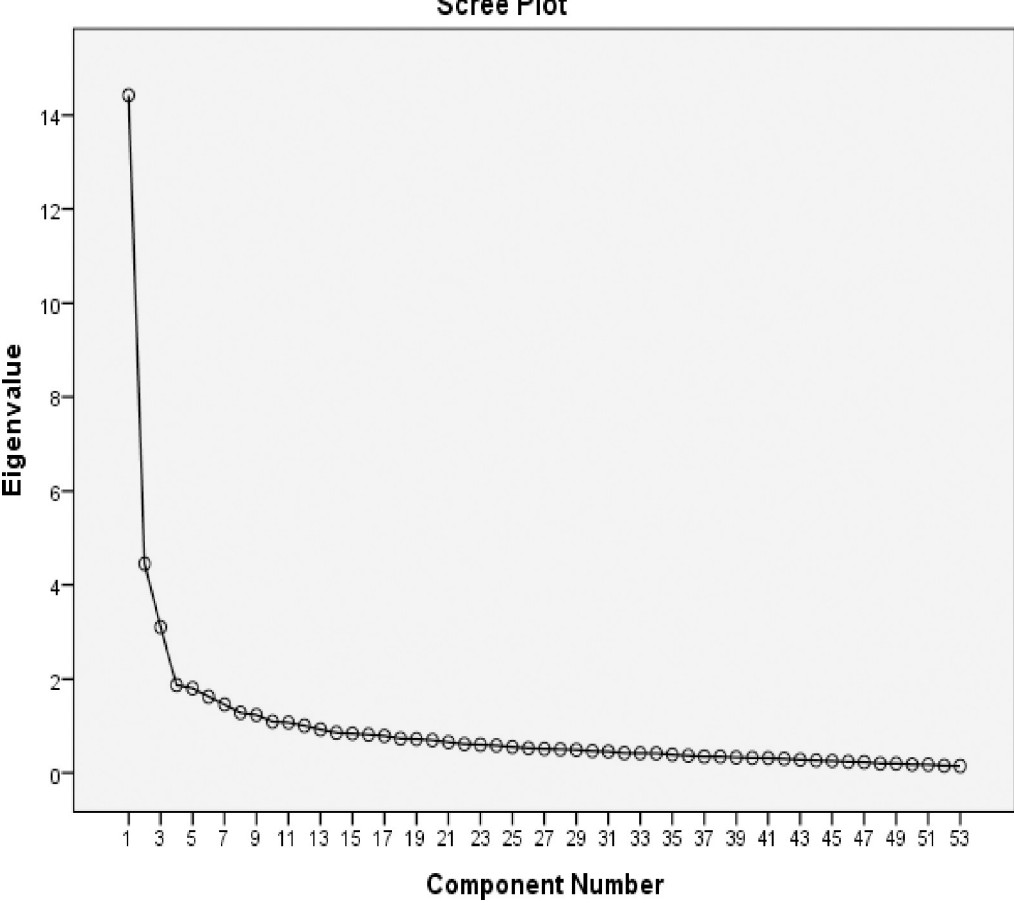

**Fig 1. Scree Plot was used to determine the number of factors, based on this chart, 4 factors were proposed for extraction in exploratory factor analysis of MHELIP questionnaire.**

## Discussion

The purpose of this study was to design and evaluate the validation indexes of a questionnaire for measuring maternal health literacy in pregnant women. The initial questionnaire was designed based on data extracted from qualitative study of pregnant women, using expert opinions and reviewing existing studies in health literacy. After completing the validity and reliability stages, a maternal health literacy questionnaire was developed with 48 questions in four dimensions, which was completed by the participants. Considering that the majority of participants completed the questionnaire in approximately 15 minutes, this tool could be easily used in screening. The findings of this study showed that, the maternal health literacy questionnaire had accepted validity and reliability.

The first subscale of the maternal health literacy tool during pregnancy (MHELIP) was a 21-item construct that measured maternal health-related knowledge (questions 1 to 21). In many maternal health literacy tools such as Chanyuan et al (2015) [35] and McCathern et al (2011), [44] there is a separate subscale called "maternal health-related knowledge", but the content and nature of questions as well as types of responses are different in these questionnaires, and they do not have the same subject diversity and comprehensiveness as our designed tool.

**Table 2. The 4-factor construct and factor load of each item.**

| Item | First factor | Second factor | Third factor | Fourth factor |
|---|---|---|---|---|
| I know natural physical changes during pregnancy. | **0.510** | 0.157 | 0.287 | 0.234 |
| I know natural psychological changes during pregnancy. | **0.521** | 0.149 | 0.282 | 0.148 |
| I know proper nutrition during pregnancy. | **0.605** | 0.216 | 0.191 | 0.059 |
| I know personal health care. | **0.599** | 0.174 | 0.319 | 0.052 |
| I know Proper activity and status in pregnancy. | **0.703** | 0.171 | 0.107 | 0.042 |
| I know proper exercise during pregnancy. | **0.590** | 0.079 | 0.187 | 0.001 |
| I know pregnancy supplements (vitamins). | **0.665** | 0.171 | 0.107 | 0.042 |
| I know the appropriate referral timing for pregnancy examinations (visits). | **0.644** | 0.168 | 0.207 | 0.025 |
| I know diagnostic examination (ultrasound and tests) of maternal and fetal health in pregnancy. | **0.707** | 0.175 | 0.143 | 0.053 |
| I know the acceptable and normal amount of weight gain during pregnancy. | **0.717** | 0.155 | 0.131 | 0.007 |
| I know common pregnancy problems such as nausea, vomiting, lower back pain. | **0.681** | 0.125 | 0.105 | 0.069 |
| I know injecting safe (allowed) vaccines during pregnancy. | **0.681** | 0.133 | 0.022 | 0.097 |
| I know the proper sexual relation during pregnancy. | **0.719** | 0.239 | 0.060 | 0.087 |
| I know the normal number of fetal movements. | **0.696** | 0.158 | 0.012 | 0.117 |
| I know the factors affecting fetal health such as photography, medications, chemicals such as botox, etc. | **0.648** | 0.101 | 0.134 | 0.105 |
| I know risk signs in pregnancy. | **0.738** | 0.102 | 0.020 | 0.109 |
| I know pregnancy disease symptoms such as gestational diabetes, high blood pressure in pregnancy and other diseases. | **0.714** | 0.074 | 0.110 | 0.215 |
| I know childbirth such as the advantages and disadvantages of each of the natural delivery methods and cesarean section and their associated care. | **0.630** | 0.097 | 0.064 | 0.238 |
| I know the methods of pain relief in virginal delivery. | **0.544** | 0.019 | 0.124 | 0.165 |
| I know neonatal and infant care in the postpartum period. | **0.596** | 0.183 | 0.143 | 0.316 |
| I know required postpartum care of mother. | **0.653** | 0.130 | 0.080 | 0.289 |
| I acquire information from written materials such as books, educational notes, pamphlets and medication brochures. | 0.102 | 0.205 | 0.263 | **0.412** |
| I acquire information from radio and television. | 0.157 | 0.046 | 0.001 | **0.467** |
| I acquire information from internet sources such as websites, instagram and telegram. | 0.152 | 0.058 | 0.061 | **0.767** |
| I acquire information from other pregnant women. | 0.016 | 0.041 | 0.199 | **0.580** |
| I acquire information from family, friends and acquaintances. | 0.068 | 0.000 | 0.033 | **0. 590** |
| I acquire information from healthcare professionals such as a physician or midwife. | 0.224 | 0.302 | 0.115 | **0.378** |
| It is easy for me to read and pronounce pregnancy-related vocabulary from information sources such as books, educational booklets, internet, telegram and Instagram. | 0.206 | 0.122 | **0.734** | 0.025 |
| The information obtained from different sources of information are understandable for me. | 0.287 | 0.296 | **0.611** | 0.032 |
| I know valid and verified sources for getting the right pregnancy related information. | 0.233 | 0.044 | **0.733** | 0.113 |
| I ask of the doctor or midwife to make sure pregnancy related information. | 0.144 | 0.205 | **0.385** | 0.293 |
| I Evaluate the accuracy of pregnancy-related information obtained from online sources such as websites, instagram and telegram. | 0.098 | 0.075 | **0.758** | 0.239 |
| I Evaluate the accuracy of pregnancy-related information obtained from friends and relatives | 0.150 | 0.214 | **0.414** | 0.084 |
| I able to control/management physical and psychological changes in pregnancy. | 0.329 | **0.539** | 0.157 | 0.205 |
| I implement a proper diet for pregnancy. | 0.244 | **0.591** | 0.030 | 0.072 |
| I implement necessary measures for personal health care during pregnancy. | 0.224 | **0.575** | 0.105 | 0.033 |
| I adhere to the principles of activity and proper condition during pregnancy. | 0.352 | **0.496** | 0.005 | 0.040 |
| I take pregnancy supplements as prescribe by doctor or midwife. | 0.101 | **0.546** | 0.044 | 0.079 |
| I consult with the doctor or midwife for taking any type of medication during pregnancy (chemical and herbal). | 0.036 | **0.615** | 0.214 | 0.086 |
| I attend for prenatal care (examinations) as scheduled. | 0.138 | **0.652** | 0.041 | 0.016 |
| I Perform ultrasound and tests in pregnancy recommended by healthcare professionals such as doctor or midwife. | 0.046 | **0.737** | 0.111 | 0.017 |
| I monitor and control the weight gain during pregnancy. | 0.079 | **0.590** | 0.053 | 0.012 |
| I use the appropriate methods of sexual relation during pregnancy. | 0.161 | **0.557** | 0.014 | 0.10 |

*(Continued)*

**Table 2.** (Continued)

| Item | First factor | Second factor | Third factor | Fourth factor |
|---|---|---|---|---|
| I avoid taking actions that are harmful to pregnancy. | 0.177 | **0.528** | 0.024 | 0.038 |
| I see the doctor or midwife as soon as possible when any signs of danger in pregnancy is observed. | 0.072 | **0.645** | 0.001 | 0.049 |
| I ask the doctor or midwife for further explanation if the information and recommendations are not clear enough. | 0.051 | **0.629** | 0.216 | 0.253 |
| I participate in decision making about pregnancy issues with the doctor or midwife (providing personal opinions). | 0.100 | **0. 476** | 0.277 | 0.207 |
| I pay attention to the accuracy and appropriateness of information given to other pregnant women. | 0.043 | **0.442** | 0.276 | 0.305 |
| Rotation sums of squared loadings | **20.73** | **33.55** | **41.50** | **46.49** |
| Percentage of data dispersion by each dimension | **20.73** | **12.81** | **7.95** | **4.99** |
| Percentage of variance coverage of total changes | **49.46** | | | |

**Factor1:** Maternal Health Knowledge, **factor2:** Maternal Health Decision Making and Behavior, **factor3:** Assessment of Maternal Health Information, **factor4:** Search for maternal health information.

The second subscale of MHELIP was the maternal health information search that consisted of 6 questions (questions 22 to 27). Although health seeking behavior is an important component of health literacy, in most health literacy questionnaires, this factor has not been considered as a separate sub-scale, even though this subscale is one of the coherent conceptual dimensions of health literacy in pregnant women according to our qualitative study.

The third subscale of MHELIP was the maternal health information assessment that included 6 items (questions 28 to 33) that were identified in exploratory factor analysis. In the Guttersrud et al (2015) questionnaire [34], a subscale called; "evaluation of health information" has been added to the maternal health literacy assessment questionnaire as an independent dimension. However, the nature of items in this questionnaire differs from our designed questionnaire.

The fourth subscale of maternal health literacy questionnaire was the maternal health decision making and behavior that included 15 items (questions 34 to 48). Pregnancy is an important period, in which the mother needs to take appropriate health-related behaviors and decision-making in order to maintain her health and the health of her fetus. There are no such subscales in other maternal health literacy questionnaires and questions that are close to ones in this section fall into the general scope of functional health literacy and are not as comprehensive as the questions in our designed tool. On the one hand, in this section of our designed tool, there are some questions about interacting with health care professionals, and having appropriate attention and behavior in regard to sharing of accurate information with other pregnant mothers, which do not exist in other maternal health literacy tools.

**Table 3. The scores of Cronbach's alpha coefficient, ICC constructs and the whole MHELIP tool.**

| No | Construct | Number of items | Cronbach's alpha coefficient | *ICC | **CI (95%) | $P_{value}$ |
|---|---|---|---|---|---|---|
| 1 | Maternal Health Knowledge | 21 | 0.94 | 0.97 | 0.927–0.990 | 0.000 |
| 2 | Search for maternal health information | 6 | 0.66 | 0.74 | 0.552–0.897 | 0.000 |
| 3 | Assessment of Maternal Health Information | 6 | 0.79 | 0.86 | 0.690–0.932 | 0.000 |
| 4 | Maternal Health Decision Making and Behavior | 15 | 0.87 | 0.92 | 0.853–0.971 | 0.000 |
| 5 | Overall reliability of the tool | 48 | 0.94 | 0.96 | 0.927–0.984 | 0.000 |

*ICC: Infraclass correlation coefficient.

**confidence intervals 95%.

Studies have suggested that health literacy is a social structure that should be considered as a multidimensional hidden construct [45]. Therefore, to measure it, we need different tools for different areas. Also, since a single definition of health literacy cannot be provided, it cannot be measured by a single tool, as the complexity of health literacy requires a multidimensional tool. The literature review showed that Chanyuan et al (2015) [35], McCathern (2011) [44], Kharazi et al (2016) [46], and Guttersrud et al (2015) [34] have designed or validated some tools to assess health literacy during pregnancy, but in the present study, items of the designed tool were extracted from a qualitative study in which, the concept of maternal health literacy was explained according to the understanding and experience of pregnant women. None of the tools available in this regard has highlighted the importance of using the experiences of pregnant women as their primary target group.

One of the most important features of a holistic tool is its comprehensiveness that covers all dimensions of a concept. In the present study, four domains extracted in the qualitative phase were also expressed during factor analysis, while in other tools the number of factors is less and even their items do not have the proper consistency with the titles of above-mentioned dimensions. On the other hand, although the concept of health literacy begins with the information seeking behavior, the subscale of maternal health information search has not been considered as a separate and independent dimension. However, in our designed tool, an independent dimension had acceptable and stable number of items and appropriate reliability.

In the designing of the tool, we tried to make the items of each domain truly genuine, have suitable comprehensiveness and be specific for pregnancy period, which were confirmed in the factor analysis. Meanwhile in tools such as the one designed by Kharrazi et al (2016) [46], the dimension of speech and listening perception includes items that are quite general, such as I can read and write, etc. In the designed questionnaire, based on the fact that mothers are seeking information about postpartum health during pregnancy, the information on postpartum health was defined in terms of mother and child health. But since the tool was intended to be a pregnancy specific tool and the questions should be able to cover the pregnancy itself, only two general items were measuring the maternal knowledge about postpartum health of mother and infant. In the Chanyuan (2015) tool [35], the dimension of knowledge and the concept of basic dimensions of maternal health included items that covered not only pregnancy but also postpartum, and in the domain of lifestyle and behavioral dimensions (which regardless of the domain title and content of items, examine maternal knowledge), only one item was related to the pregnancy and the rest were related to the postpartum period. Also, in their tool the dimension of basic skills during pregnancy examines the ability to read a book, attend a pregnancy class, have a scheduled appointment, and have at least five visits to prenatal clinic, which only the item of "I regularly attend the scheduled visits" has been designed as one of the items of the questionnaire. This is while the decision-making and behavior construct of the designed questionnaire examines the health literacy of the individual in this dimension in a broader way and this leads to the better understanding of individual's situation.

In the Guttersrud et al (2015) questionnaire [34], the domain of one's perception of competence and adaptation skills includes questions that examine information search, information comprehension, ability to transfer information to others, ability to recognize the risk symptoms, follow-up and readiness for childbirth, which are similar to the assessment, decision making and behavior, and the knowledge sections of our designed questionnaire. However, despite this similarity, the nature of questions in the two tools is not the same. In the Guttersrud et al (2015) tool [34], the information assessment questions include separation of right from wrong, easy recall of previous information, independence in following recommendations, being active, social participation like pre-pregnancy period, and ability to care for own self and the infant. The questions in the above tool are a mixture of questions on various

aspects of health literacy and it is true that the title of this construct is similar to our designed tool, but the nature of its questions are different from the tool designed in the present study. Perhaps the concept of health literacy is also different in the two tools. This is the strength of our designed tool that reports health literacy as a whole, and also as dimensions that are completely independent of one another, which can confirm the importance of a mixed exploratory study that could complete the entire psychometric evaluation process, including construct validity.

The tool designed to measure pregnancy knowledge covers issues needed by pregnant women. Questions in this tool are suitable for the general public of pregnant mothers while in other tools such as Chanyuan (2015) tool [35], Knowledge questions may not have a high difficulty coefficient and low discriminate coefficient for identifying individuals with different levels of health literacy. But, this difference in the design of questions may be due to the knowledge of researcher about the community of pregnant woman and his/her expectation about the knowledge of pregnant women in that community. Also, the prenatal care questionnaire used by McCathern [44] (2011) had 5 items with 4-option answer in the field of prenatal care knowledge, and although this type of questions can measure the level of knowledge properly, it may be possible to ask if these tools were qualitatively faced validated by pregnant mothers, or would they still obtain an acceptable score of importance? Failure to perform quantitative face validity by those who are the most important target group of the questionnaire is one of the weaknesses of this questionnaire and similar questionnaires that have been considered in the designed tool. The main strengths of present study is that, it is the first fundamental step in examining the health literacy of Iranian pregnant women and also this tool can be an incentive to initiate extensive and substantive research on health literacy in pregnant women. The most important application of this tool is that, it can be used as a screening tool in prenatal care settings to measure the general maternal health literacy level and its various subsets so that, subsequent planning can be made for this vulnerable group. One of the main limitations of this study is that, it was conducted in an urban area and may not represent the general population of pregnant women. Other researchers are suggested to examine the confirmatory factor analysis of our designed tool and validate the MHELIP questionnaire in other parts of Iran and also other countries to address its potential weakness. It is recommended to validate the MHELIP questionnaire in two groups of nulliparous and multiparous pregnant women separately.

## Conclusion

The maternal health literacy assessment tool during pregnancy is a valid and reliable tool and can be used in future studies to measure pregnant women's health literacy.

## Supporting information

**S1 File.**
(DOCX)

**S2 File. Questions guide in qualitative phase.**
(DOCX)

**S3 File.**
(SAV)

**S4 File. The maternal health literacy inventory in pregnancy (MHELIP).**
(DOCX)

**S5 File. Manual for scoring the maternal health literacy inventory in pregnancy (MHELIP).** (DOC)

## Acknowledgments

This study is part of a PhD dissertation at Tehran University of Medical Sciences with code of ethics: IR.TUMS.VCR.REC.1395.1866. We would like to thank and appreciate all the experts and participants who helped us with this study.

## Author Contributions

**Conceptualization:** Safoura Taheri, Ziba Taghizadeh.

**Data curation:** Safoura Taheri, Zohre Momenimovahed, Azita Tiznobaik, Ziba Taghizadeh.

**Formal analysis:** Safoura Taheri, Zohre Momenimovahed, Azita Tiznobaik, Ziba Taghizadeh.

**Funding acquisition:** Safoura Taheri, Ziba Taghizadeh.

**Investigation:** Safoura Taheri, Ziba Taghizadeh.

**Methodology:** Safoura Taheri, Mahmoud Tavousi, Ziba Taghizadeh.

**Project administration:** Safoura Taheri, Ziba Taghizadeh.

**Resources:** Safoura Taheri, Ziba Taghizadeh.

**Software:** Safoura Taheri, Ziba Taghizadeh.

**Supervision:** Safoura Taheri, Mahmoud Tavousi, Ziba Taghizadeh.

**Validation:** Safoura Taheri, Mahmoud Tavousi, Ziba Taghizadeh.

**Visualization:** Safoura Taheri, Ziba Taghizadeh.

**Writing – original draft:** Safoura Taheri, Ashraf Direkvand-Moghadam, Zainab Suhrabi, Ziba Taghizadeh.

**Writing – review & editing:** Safoura Taheri, Ashraf Direkvand-Moghadam, Zainab Suhrabi, Ziba Taghizadeh.

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
