## [Decision Letter · Decision Letter 0]

10 Dec 2019

PONE-D-19-24536

Development and Psychometric Properties of maternal health literacy inventory in pregnancy (MHELIP)

PLOS ONE

Dear Dr taghizadeh,

Thank you for submitting your manuscript to PLOS ONE. After careful consideration, we feel that it has merit but does not fully meet PLOS ONE’s publication criteria as it currently stands. Therefore, we invite you to submit a revised version of the manuscript that addresses the points raised during the review process.

We would appreciate receiving your revised manuscript by Jan 24 2020 11:59PM. To enhance the reproducibility of your results, we recommend that if applicable you deposit your laboratory protocols in protocols.io, where a protocol can be assigned its own identifier (DOI) such that it can be cited independently in the future. For instructions see: http://journals.plos.org/plosone/s/submission-guidelines#loc-laboratory-protocols

We look forward to receiving your revised manuscript.

Kind regards,

Wen-Jun Tu

Academic Editor

PLOS ONE

Journal Requirements:

2) Please include additional information regarding the survey or questionnaire used in the study and ensure that you have provided sufficient details that others could replicate the analyses. For instance, if you developed a questionnaire as part of this study and it is not under a copyright more restrictive than CC-BY, please include a copy, in both the original language and English, as Supporting Information.

3) We suggest you thoroughly copyedit your manuscript for language usage, spelling, and grammar. If you do not know anyone who can help you do this, you may wish to consider employing a professional scientific editing service.  

4) Please confirm in your methods section and ethics statement that the 'Faculty of Nursing and Midwifery of Tehran University of Medical Sciences' consists of an Institutional Review Board (IRB) or ethics committee of experts that reviewed and approved your study.

In addition, please provide additional details regarding participant consent. In the ethics statement in the Methods and online submission information, please ensure that you have specified whether consent was informed.

5) Thank you for stating the following financial disclosure:

Please provide an amended Funding Statement that declares *all* the funding or sources of support received during this specific study (whether external or internal to your organization) as detailed online in our guide for authors at http://journals.plos.org/plosone/s/submit-now.  Please state what role the funders took in the study.  If any authors received a salary from any of your funders, please state which authors and which funder. If the funders had no role, please state: "The funders had no role in study design, data collection and analysis, decision to publish, or preparation of the manuscript."

6) We note that you have indicated that data from this study are available upon request. PLOS only allows data to be available upon request if there are legal or ethical restrictions on sharing data publicly. For more information on unacceptable data access restrictions, please see http://journals.plos.org/plosone/s/data-availability#loc-unacceptable-data-access-restrictions.

7) PLOS requires an ORCID iD for the corresponding author in Editorial Manager on papers submitted after December 6th, 2016. Please ensure that you have an ORCID iD and that it is validated in Editorial Manager. To do this, go to ‘Update my Information’ (in the upper left-hand corner of the main menu), and click on the Fetch/Validate link next to the ORCID field. This will take you to the ORCID site and allow you to create a new iD or authenticate a pre-existing iD in Editorial Manager. Please see the following video for instructions on linking an ORCID iD to your Editorial Manager account: https://www.youtube.com/watch?v=_xcclfuvtxQ

8)  Please upload a copy of Figures 2-4, to which you refer in your text on page 15. If any figure is no longer to be included as part of the submission please remove all reference to it within the text.

9) Please include a separate caption for each figure in your manuscript.

Reviewers' comments:

Reviewer's Responses to Questions

**Comments to the Author**

1. Is the manuscript technically sound, and do the data support the conclusions?

Reviewer #1: Yes

Reviewer #2: Yes

2. Has the statistical analysis been performed appropriately and rigorously? 

Reviewer #1: Yes

Reviewer #2: Yes

3. Have the authors made all data underlying the findings in their manuscript fully available?

Reviewer #1: Yes

Reviewer #2: No

4. Is the manuscript presented in an intelligible fashion and written in standard English?

Reviewer #1: Yes

Reviewer #2: Yes

5. Review Comments to the Author

Reviewer #1: Please check the language of items once again. See the attachment for my specific review comments. The article is technically sound and has merit for publication. I recommend for acceptance with minor correction. However, it is better that the whole manuscript is proofread by a professional preferably by a native English speaker.

Reviewer #2: The aim of this study was to develop a tool to measure maternal health literacy during pregnancy and test its psychometric properties. This is an interesting and timely study particularly in the low and middle-income country context. I have few comments for the authors to consider:

1.I would suggest authors to rephrase their study objective mentioned in the abstract (perhaps in the introduction too).

2.In the last paragraph of the introduction, authors have mentioned: “but a native tool derived from a genuine research”, I would encourage authors to replace the word ‘genuine’ from ‘original’ or another appropriate word.

3.When was this study actually started- November 2016 (mentioned in material and methods) / February 2016 (mentioned in qualitative study)? Please correct the inconsistency.

4.How did authors actually recruit participants (19 pregnant women) for the qualitative study (over 1.5+ years)? More details are needed.

5.I suggest authors to state the number of semi-structured and in-depth interviews conducted separately.

6.How did authors conduct 23 individual interviews with 19 pregnant women?

7.Please provide your interview guide as an appendix.

8.In my opinion, below chunk of text needs to be in the result section.

After analyzing the findings of qualitative section and reviewing the available literature and tools, a 124-item pool was created. After reviewing the items, the members of primary research group developed a 78-item Maternal Health Literacy Assessment Tool (MHELIP) questionnaire, consisting of two sections: 1) Assessing the information related to pregnancy health, and 2) Functional health literacy. For the initial scoring, a 5-point Likert scale was used in the section of assessing the information related to pregnancy health, which ranged from I don't know at all to I know a lot (with the score of 1 to 5), and a scale of never to always (with the score of 1 to 5) was used in the section of functional health literacy.

9.How did authors compute Content Validity Ratio (CVR) and Content Validity Index (CVI)?

10.I suggest authors to expand on the ‘impact score method’ in the face validity section.

11.Please provide references for Content Validity Ratio (CVR) and Content Validity Index (CVI), Lawche’s table and impact score method.

12.How did authors collect data for psychometric analysis, e.g. sample selection, who collected the data, setting, etc., and please explain?

13.Perhaps, authors could also use a flow chart to summarize/simplify the entire psychometric evaluation process and scale item selection.

14.How did authors identify participants having minimum level of education for reading and writing?

15.Eligible participants who are not willing to participate to a study are generally called as non-respondents. However, authors have listed them under the exclusion criteria here. In this case, there is a possibility of having a systematic difference between respondents and non-respondents.

16.I suggest to present participant characteristics of the qualitative study and then consider addressing reviewer comment 8.

17.Please revisit your interpretation on Bartlett’s test of sphericity. https://www.ibm.com/support/knowledgecenter/SSLVMB_26.0.0/statistics_casestudies_project_ddita/spss/tutorials/fac_telco_kmo_01.html

18.There is an inconsistency of figure labelling [e.g. Figure 1-4 (in text)]

19.Suggest to report Cronbach’s alpha and ICC values with 95% confidence intervals.

20.Please highlight the main strengths and limitations of this study.

21.Please correct the typos and inconsistencies in the reference section, e.g. no 5.

6. PLOS authors have the option to publish the peer review history of their article (what does this mean?). If published, this will include your full peer review and any attached files.

Reviewer #1: No

Reviewer #2: Yes: Dhammika Deepani Siriwardhana

---

## [Author Response · Author response to Decision Letter 0]

17 Mar 2020

The aim of this study was to develop a tool to measure maternal health literacy during pregnancy and test its psychometric properties. This is an interesting and timely study particularly in the low and middle-income country context. I have few comments for the authors to consider: 

dear reviewer, thank you for your valuable comments.

1. I would suggest authors to rephrase their study objective mentioned in the abstract (perhaps in the introduction too). Applied to the article. 33-35 line, 114-116 line.

2. In the last paragraph of the introduction, authors have mentioned: “but a native tool derived from a genuine research”, I would encourage authors to replace the word ‘genuine’ from ‘original’ or another appropriate word. Applied to the article. 108 line

3. When was this study actually started- November 2016 (mentioned in material and methods) / February 2016 (mentioned in qualitative study)? Please correct the inconsistency. Applied to the article.128 line and 150-153 line.

4. How did authors actually recruit participants (19 pregnant women) for the qualitative study (over 1.5+ years)? More details are needed. Applied to the article.128line

The first phase (qualitative study) of the study was performed from February 2016 to September 2017, about Seven- Eight months.

5. I suggest authors to state the number of semi-structured and in-depth interviews conducted separately. Applied to the article.128-131 line., 135-137 line

6. How did authors conduct 23 individual interviews with 19 pregnant women? Applied to the article135-137 line

7. Please provide your interview guide as an appendix. Applied in support information

8. In my opinion, below chunk of text needs to be in the result section. Applied to the article. 210-218 line

After analyzing the findings of qualitative section and reviewing the available literature and tools, a 124-item pool was created. After reviewing the items, the members of primary research group developed a 78-item Maternal Health Literacy Assessment Tool (MHELIP) questionnaire, consisting of two sections: 1) Assessing the information related to pregnancy health, and 2) Functional health literacy. For the initial scoring, a 5-point Likert scale was used in the section of assessing the information related to pregnancy health, which ranged from I don't know at all to I know a lot (with the score of 1 to 5), and a scale of never to always (with the score of 1 to 5) was used in the section of functional health literacy.

9. How did authors compute Content Validity Ratio (CVR) and Content Validity Index (CVI)? Applied to the article. 180-190 line

10. I suggest authors to expand on the ‘impact score method’ in the face validity section. Applied to the article. 170-175 line

11. Please provide references for Content Validity Ratio (CVR) and Content Validity Index (CVI), Lawche’s table and impact score method. Applied to the article. 163-166 line

12. How did authors collect data for psychometric analysis, e.g. sample selection, who collected the data, setting, etc., and please explain? Applied to the article. 180-190 line

13. Perhaps, authors could also use a flow chart to summarize/simplify the entire psychometric evaluation process and scale item selection. Applied to the article.178 and 558 line.

How did authors identify participants having minimum level of education for reading and writing? Applied to the article. 178line. When, the samples invited to the study, the researcher asked them about Ability to read and write.

14. Eligible participants who are not willing to participate to a study are generally called as non-respondents. However, authors have listed them under the exclusion criteria here. In this case, there is a possibility of having a systematic difference between respondents and non-respondents. Applied to the article. First 189line. The sentence changed :Incomplete questionnaire completion

15. I suggest to present participant characteristics of the qualitative study and then consider addressing reviewer comment 8. Applied to the article. 207-218 line

16. Please revisit your interpretation on Bartlett’s test of sphericity. https://www.ibm.com/support/knowledgecenter/SSLVMB_26.0.0/statistics_casestudies_project_ddita/spss/tutorials/fac_telco_kmo_01.html. Applied to the article.244-245 line

17. There is an inconsistency of figure labelling [e.g. Figure 1-4 (in text)] Applied to the article. 248 line.

18. Suggest to report Cronbach’s alpha and ICC values with 95% confidence intervals. Applied to the article. 264-268 line

19. Please highlight the main strengths and limitations of this study. Applied to the article. 374-385line

20. Please correct the typos and inconsistencies in the reference section, e.g. no 5. Applied to the article. 413-548 line

---

## [Decision Letter · Decision Letter 1]

17 Apr 2020

PONE-D-19-24536R1

Development and Psychometric Properties of maternal health literacy inventory in pregnancy (MHELIP)

PLOS ONE

Dear Dr taghizadeh,

Thank you for submitting your manuscript to PLOS ONE. After careful consideration, we feel that it has merit but does not fully meet PLOS ONE’s publication criteria as it currently stands. Therefore, we invite you to submit a revised version of the manuscript that addresses the points raised during the review process.

We would appreciate receiving your revised manuscript by Jun 01 2020 11:59PM. To enhance the reproducibility of your results, we recommend that if applicable you deposit your laboratory protocols in protocols.io, where a protocol can be assigned its own identifier (DOI) such that it can be cited independently in the future. For instructions see: http://journals.plos.org/plosone/s/submission-guidelines#loc-laboratory-protocols

We look forward to receiving your revised manuscript.

Kind regards,

Wen-Jun Tu

Academic Editor

PLOS ONE

Reviewers' comments:

Reviewer's Responses to Questions

**Comments to the Author**

1. If the authors have adequately addressed your comments raised in a previous round of review and you feel that this manuscript is now acceptable for publication, you may indicate that here to bypass the “Comments to the Author” section, enter your conflict of interest statement in the “Confidential to Editor” section, and submit your "Accept" recommendation.

Reviewer #1: All comments have been addressed

Reviewer #2: (No Response)

2. Is the manuscript technically sound, and do the data support the conclusions?

Reviewer #1: Yes

Reviewer #2: Yes

3. Has the statistical analysis been performed appropriately and rigorously? 

Reviewer #1: Yes

Reviewer #2: Yes

4. Have the authors made all data underlying the findings in their manuscript fully available?

Reviewer #1: Yes

Reviewer #2: Yes

5. Is the manuscript presented in an intelligible fashion and written in standard English?

Reviewer #1: Yes

Reviewer #2: No

6. Review Comments to the Author

Reviewer #1: The manuscript has been revised accordingly. It has good merit for publication. I recommend for acceptance.

Reviewer #2: Line 124, I suggest the authors to mention their study period as February 2016 to August 2018.

Line 140-142-did the authors exclude four interviews (first round) during the analysis since they have repeated interviews with four women? Please clarify.

5. I suggest authors to state the number of semi-structured and in-depth interviews

conducted separately. Applied to the article.128-131 line., 135-137 line

No information is provided on the number of semi-structured interviews and in-depth interviews conducted. In the abstract authors have only mentioned about in-depth interviews.

18. Suggest to report Cronbach’s alpha and ICC values with 95% confidence intervals.

Applied to the article. 264-268 line

This comment has been addressed by the authors incorrectly.

In the manuscript text and Table 3, 95% confidence intervals are not presented for Cronbach’s alpha and ICC values. Please mention the values.

20. Please correct the typos and inconsistencies in the reference section, e.g. no 5. Applied

to the article. 413-548 line

p value of Bartlett’s test should be written as follows: (p value <0.001).

Suggest to revise the manuscript with a native English speaker.

7. PLOS authors have the option to publish the peer review history of their article (what does this mean?). If published, this will include your full peer review and any attached files.

Reviewer #1: Yes: Md. Sabiruzzaman

Reviewer #2: Yes: Dhammika Deepani Siriwardhana

---

## [Author Response · Author response to Decision Letter 1]

7 May 2020

dear reviewer, thank you for your valuable comments

Reviewer #1: The manuscript has been revised accordingly. It has good merit for publication. I recommend for acceptance. Thanks a lot.

Reviewer #2: Line 124, I suggest the authors to mention their study period as February 2016 to August 2018. Applied to the article. 120 line

Line 140-142-did the authors exclude four interviews (first round) during the analysis since they have repeated interviews with four women? Please clarify. Applied to the article. 131-132 line. In total, we interviewed 19 pregnant mothers individually, deeply and semi-structured. In 4 of these individuals (19 pregnant) the interview lasted and at another time the interview continued to complete the accuracy. The sentence was corrected to clear up the ambiguity

5. I suggest authors to state the number of semi-structured and in-depth interviews

conducted separately. Applied to the article.128-131 line., 135-137 line. 

No information is provided on the number of semi-structured interviews and in-depth interviews conducted. In the abstract authors have only mentioned about in-depth interviews. Applied to the article. 38-39 line 131-132 line. In total, we interviewed 19 pregnant mothers individually, deeply and semi-structured. In 4 of these individuals (19 pregnant) the interview lasted and at another time the interview continued to complete the accuracy. The sentence was corrected to clear up the ambiguity

18. Suggest to report Cronbach’s alpha and ICC values with 95% confidence intervals.

Applied to the article. 264-268 line

This comment has been addressed by the authors incorrectly.

In the manuscript text and Table 3, 95% confidence intervals are not presented for Cronbach’s alpha and ICC values. Please mention the values. Applied to the article. 597 line

20. Please correct the typos and inconsistencies in the reference section, e.g. no 5. Applied

to the article. 413-548 line . Applied to the article. 424-559 line

p value of Bartlett’s test should be written as follows: (p value <0.001). Applied to the article256 line

Suggest to revise the manuscript with a native English speaker. Edited by Native

---

## [Editor Report · Decision Letter 2]

26 May 2020

Development and psychometric properties of maternal health literacy inventory in pregnancy

PONE-D-19-24536R2

Dear Dr. taghizadeh,

We are pleased to inform you that your manuscript has been judged scientifically suitable for publication and will be formally accepted for publication once it complies with all outstanding technical requirements.

With kind regards,

Wen-Jun Tu

Academic Editor

PLOS ONE
---

## [Editor Report · Acceptance letter]

3 Jun 2020

PONE-D-19-24536R2 

Development and psychometric properties of maternal health literacy inventory in pregnancy 

Dear Dr. taghizadeh:

I'm pleased to inform you that your manuscript has been deemed suitable for publication in PLOS ONE. Congratulations! Your manuscript is now with our production department. 

Kind regards, 

on behalf of

Dr. Wen-Jun Tu 

Academic Editor

PLOS ONE